# On the driver of blood circulation beyond the heart

Zheng Li *, Gerald H. Pollack *

Department of Bioengineering, University of Washington, Seattle, Washington, United States of America

* zhl@u.washington.edu (ZL); ghp@u.washington.edu (GHP)

## Abstract

The heart is widely acknowledged as the unique driver of blood circulation. Recently, we discovered a flow-driving mechanism that can operate without imposed pressure, using infrared (IR) energy to propel flow. We considered the possibility that, by exploiting this mechanism, blood vessels, themselves, could propel flow. We verified the existence of this driving mechanism by using a three-day-old chick-embryo model. When the heart was stopped, blood continued to flow for approximately 50 minutes, albeit at a lower velocity. When IR was introduced, the postmortem flow increased from ~41.1 ± 25.6 μm/s to ~153.0 ± 59.5 μm/s (n = 6). When IR energy was diminished under otherwise physiological conditions, blood failed to flow. Hence, this IR-dependent, vessel-based flow-driving mechanism may indeed operate in the circulatory system, complementing the action of the heart.

## Introduction

Of all physiological functions, blood circulation stands as one of the most fundamental. Few cells can survive without proper circulation. Indeed, the significance of vascular flow is difficult to overstate [1].

The sole driver of blood flow is widely acknowledged to be pressure, mainly generated from the heart [2,3]. This paradigm is arguably one of the best known in biology, seemingly leaving little room for doubt.

On the other hand, blood can apparently flow without a beating heart. After the heart had been arrested, postmortem blood flow was confirmed in mice, rats, dogs, and chick embryos [4–7]. The flow persisted from 15 minutes to several hours. Furthermore, some amphibian larvae could live up to 15 days, and even differentiate following surgical removal of the heart [8–10], implying an alternative means for propelling blood.

Continued blood flow without a beating heart raises an obvious question: could the heart be the sole driver of the circulation? Puzzled by clinical and experimental evidence that does not fit the current paradigm, generations of established physicians and physiologists since the 19[th] century have repeatedly raised this question [4,11–17]. Among those skeptics, the consensus is clear: the heart cannot be the only driver of the circulation; a complementary driving mechanism must exist in blood vessels, presumably in the capillaries [4,12–16]. However, the precise mechanism has remained unclear. Under the standard pressure-driven flow paradigm, it is not obvious how blood vessels, especially capillaries, could drive blood by themselves [4,12–16].

**Data Availability Statement:** All relevant data are within the manuscript and its Supporting Information files.

**Funding:** This work was supported by an NIH Transformative grant 5R01GM093842, received by

GHP. https://www.nih.gov/ This work was also supported by private support, received by GHP. No websites available. The funders had no role in study design, data collection and analysis, decision to publish, or preparation of the manuscript. a) None of the research costs or authors' salaries were funded, in whole or in part, by a tobacco company. b) The donor has no competing interests in relation to this work. c) The identity of the donor is not considered relevant to editor's or reviewers' assessment of the validity of the work. The authors are not aware of any competing interests.

**Competing interests:** The authors have declared that no competing interests exist.

On the other hand, pressure is not the only way to drive fluid flow through a vessel. We recently discovered that flow can also arise from a driving force involving tubular surfaces. Surface activity can generate axial chemical concentration gradients, which can drive intratubular flow [18–20]. Relative to pressure-driven flow, surface-induced flow has several distinct features, the most important of which may be its signature feature: the flow can be fueled by infrared radiation (IR) energy [20].

Blood vessels, especially capillaries, have abundant surface activities. Thus, the surface-induced flow is a natural candidate to be employed by the blood vessels to drive blood flow. Here, in an avian circulatory model, we confirm several predictions arising from this flow mechanism. First, flow continues after cardiac ejection has been stopped, following a predictable direction. Second, IR energy fuels this flow, both in the post-mortem situation and also in the normal physiological state. All of this demonstrates that other than the heart, the blood vessels can drive blood flow, too, via the surface-induced flow mechanism.

## Materials and methods

### Model selection

As a model, we employed the vitelline network of the early-stage chick embryo (Fig 1). The model has several advantages that enabled us to study the blood flow without a pressure gradient. The vascular network, diameter ~3 cm, floats on the top of the yolk, forming an almost 2D-plane. The lack of any prominent 3D feature to the vascular network has the advantage of being able to exclude gravitational force as a driving factor [21]. Further, the vascular network's small size facilitates the tracking of any potential vascular contractions [6]. This embryonic model contains all the essential elements of the circulation: the heart, blood vessels and blood; yet, the smooth muscles are not fully functional at this early stage [22–25], which can help exclude vasomotor actions as a factor when studying the flow.

### Onstage incubator

An onstage incubator was used to maintain the temperature of the egg during the experiment. The incubator was made by wrapping a flexible strip-heating element (TEMPCO, SHS80389) around an aluminum tube (inner diameter: 4.5 cm; outer diameter: 5.1 cm; height: 6.5 cm). The heating element heated the aluminum tube, while the tube's thermal radiation heated the egg. The power of the heating element, which determined the heating rate, was controlled by a power controller (Payne, 18TBP-1-15). The temperature of the heating element was controlled by a temperature controller (AGPtek, STC-1000). A temperature sensor read the stage temperature and provided feedback to the temperature controller. The temperature of the incubator was maintained at $37 \pm 0.5°C$ throughout the experiment.

### Embryo model preparation

Pathogen-free white leghorn chicken eggs (Charles River, specific pathogen-free, research grade, fertile, 55–59 grams) were incubated for 72 to 75 hours (corresponding to Hamburger–Hamilton stage 18–20 [26]) in a commercial egg incubator (Brinsea Mini Advance Hatching Egg Incubator). The temperature of the incubator was set at 37.5°C and the humidity of the incubator was maintained by continuously evaporating water in a built-in water reservoir. An integrated motor of the incubator rotated the assembly of eggs 15 degrees every 45 minutes.

After incubation, the embryo was ready for experiments. The egg was transferred to the onstage incubator in an upright position with the blunt end up. Extra padding was added at the bottom of the egg to ensure stability, and to adjust the height of the egg. A portion of the

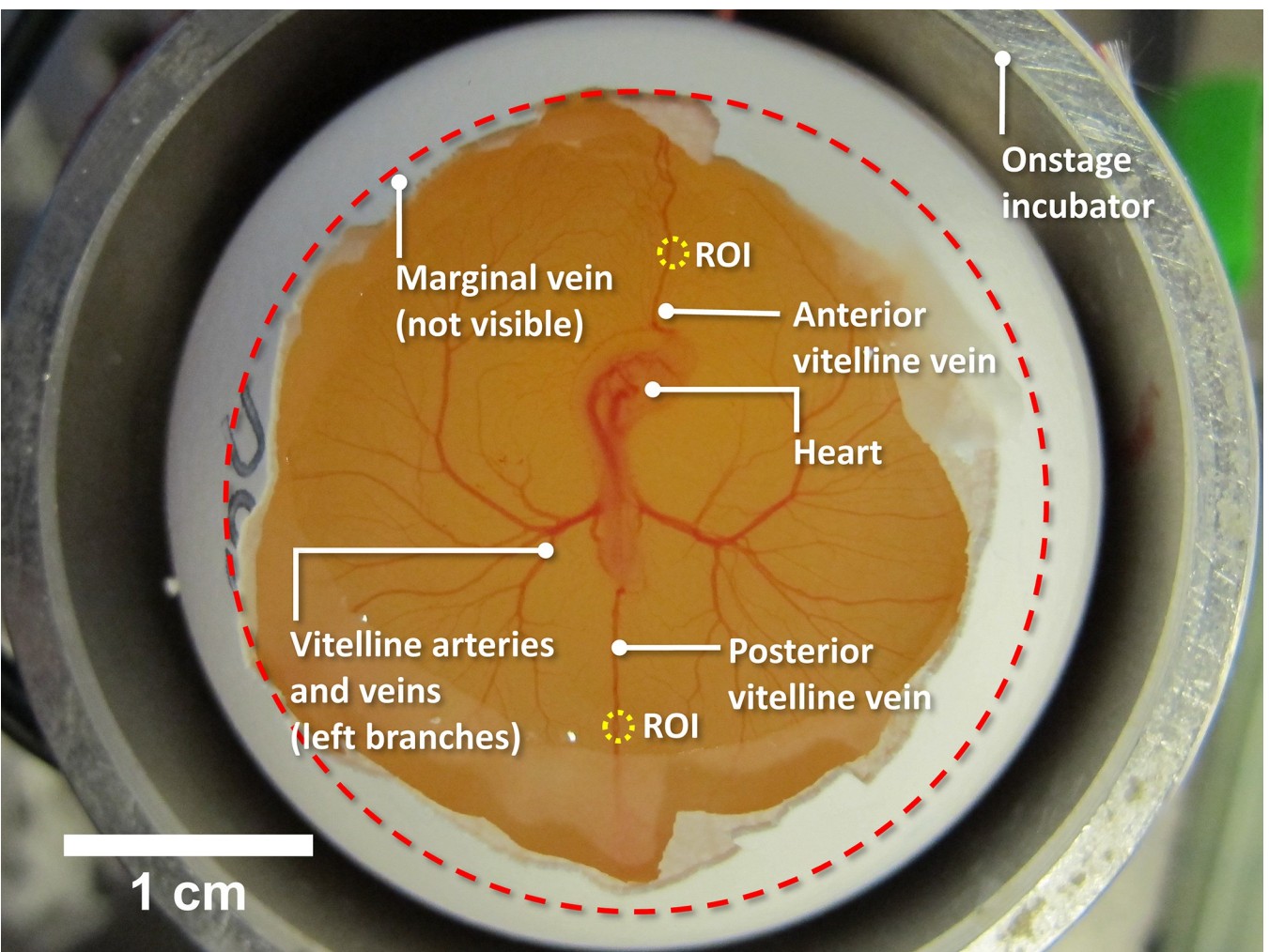

**Fig 1. Experimental setup, involving three-day old chick embryo, showing the vitelline vascular network.** The egg was situated in an onstage incubator. The shell's blunt end and the associated membrane were removed. The embryo lies on the top-middle of the egg yolk; vitelline arteries and veins grow out of the embryo, and connect to the marginal vein, whose approximate location is marked with the dashed red circle. The marginal vein forms the external boundary of the network. Anterior and posterior vitelline veins connect the marginal vein with the embryo. Because of the veins' non-branching nature and relatively uniform diameter, the midpoints of either the anterior or posterior vitelline vein, marked with yellow circles, were selected as regions of interest (ROI) for quantitative blood flow measurements. Pilot experiments showed that the blood flow velocities were similar in the two veins; hence, vein selection was based primarily on accessibility.

eggshell and the eggshell membrane were carefully removed by using a pair of tweezers (Dumont tweezer, Style 55, Domostar, 72707–01) to expose the embryo.

For post-mortem experiments, the embryonic heart was arrested by directly injecting 10–30 μl 3 M potassium chloride (J.T. Baker, 3040–01) solution into the heart with a syringe outfitted with a 27G × ½ inch needle (BD, 309623) [7,27]. Occasionally, the intrusive nature of this procedure caused cardiac hemorrhaging; those specimens with hemorrhage were excluded from the study.

## Flow data acquisition and quantification

The embryo was illuminated with white light from an LED flashlight (CREE, XLAMP XP-E LED). Blood flow was recorded using a video camera (Edmund Optics, EO-3112C) outfitted

with an objective lens (Ernst Leitz Weltzar, 6x, N.A. = 0.18 and Bausch & Lomb Opt. CO., 10x, N.A. = 0.25 for qualitative studies; Leitz Wetzlar, 32x, N.A. = 0.40 for quantitative studies). The lens was mounted onto the camera by a C-Mount to a DIN adapter (Edmund Optics, #03–627). The camera was connected to a computer via a USB interface. By decreasing the camera ROI through camera-software control, the frame rate could reach 80 frames per second.

With red blood cells as tracers, the flow video could be analyzed by Particle Image Velocimetry (PIV) to obtain red blood cell velocity [28–31]. Due to changes of blood-stream width, the flow velocities were routinely measured at the middle section of the vessels, where red blood cells were always available for tracking.

### Dynamics of postmortem blood flow

We recorded the dynamics of postmortem venous blood flow with the 32x objective lens, and recorded the postmortem arterial flow and postmortem microcirculation with the 6x or 10x objective lens instead of the 32x lens. During these experiments, the chick embryo was situated in the onstage incubator, whose temperature was maintained at $37 \pm 0.5°C$.

### Effect of IR on postmortem & physiological blood flow

With the 32x objective lens, we studied the impact of IR energy on both postmortem and physiological blood flow quantitively. Blood flow in the region of interest (ROI) (Fig 1) was recorded and analyzed.

To apply infrared radiation to the chick embryo, a 150-watt ceramic emitter (Zoo Med, CE150) was used. This lamp emitted IR at wavelengths ranging from 1 μm to 14 μm, with a peak at 2.9 μm.

In all experiments the IR source was turned on for 30 minutes prior to experimentation to stabilize the output. The IR lamp was fixed to a claw holder, with adjustable height and angle. This source was directed down to the surface of the exposed chick embryo at an angle of 45˚ with respect to the horizontal surface. The distance between the center of the IR source and the center of the embryo surface was 10 cm (Fig 2). The height and angle of the IR lamp was adjusted beforehand; and the location of the claw holder was marked on the table with tape. Thus, the IR lamp could be quickly applied and removed as needed.

After the embryo was prepared for study, the first three minutes of measurement were taken as the control. For postmortem blood-flow studies, the starting point of that control was the time that the KCl was injected into the heart. For physiological flow studies, the starting point of the control was when the embryo was suitably exposed. Immediately following the control measurement, IR was shone onto the embryo for another three minutes, during which measurement continued. Then the IR source was withdrawn, and measurement continued for the final three minutes. In all experiments, the chick embryo was continuously situated in the onstage incubator, whose temperature was maintained at $37 \pm 0.5°C$.

### Effect of IR deficiency on physiological blood flow

For physiological circulation experiments, we also observed the effect of IR deficiency. To carry out those experiments, the entire contents of the egg, including the live embryo, were poured into a 50-ml beaker (Fisher, fb-100-50), exposed to room temperature, which ranged between 19˚C and 23˚C. This configuration allowed visualization of the boundary portion of the embryo's blood vessel network, which was ordinarily blocked by the eggshell. Observations were made 1 minute and 50 minutes, respectively, after the embryo remained at room temperature.

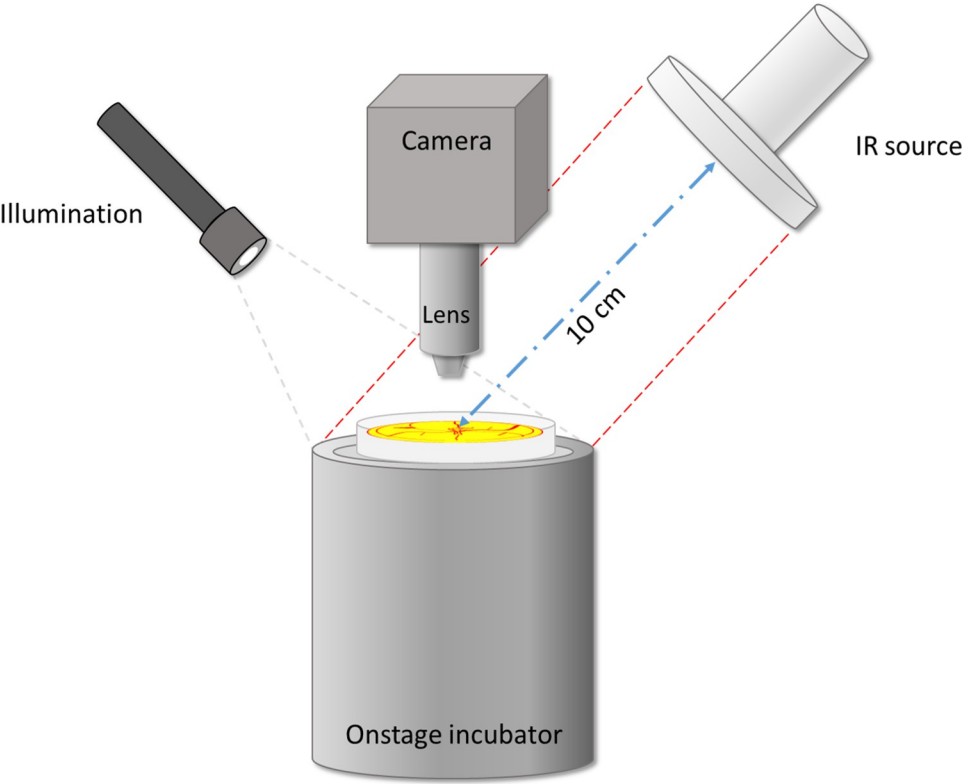

**Fig 2. Experimental configuration for applying IR to the chick embryo.**

## Results

### Dynamics of postmortem blood flow

We first observed the dynamics of postmortem blood flow. The term "postmortem" is here defined as the period after the heart had stopped beating. Within one or two seconds following KCl injection, the embryonic heart stopped. Blood flow, as measured in the ROI, persisted, although it slowed substantially (S1 Movie). Sometimes that flow even briefly reversed (S2 Movie). Following such initial stall, or reversal, venous flow slightly accelerated in the normal physiological direction, stabilizing at ~25 μm/s approximately two minutes after cardiac arrest. As the venous postmortem blood flow continued, the visualized bloodstream gradually shrunk in diameter, and the density of the red blood cells decreased compared to the physiological state. Postmortem venous flow persisted for approximately 50 minutes (S3 Movie), in a direction the same as the natural one, e.g., artery to the vein. Postmortem venous blood flow in the normal physiological direction was observed in all experiments, without exception.

In capillaries, following cardiac arrest, blood flow persisted in some beds. A qualitative impression was that less than half of the total capillary beds remained open. In those open capillaries, the flow direction was generally normal, from artery to vein (S4 Movie).

In arteries, postmortem blood flow initially reversed direction: blood flowed from thinner arterial branches back to the larger arteries. In the trunk vitelline arteries adjacent to the heart, the reversal persisted for approximately 10 minutes, longer than the brief reversal sometimes observed in veins. The restoration of natural flow direction occurred first at the peripheral region of the arterial network, gradually moving upstream and ultimately reaching the heart region (S5 Movie). Eventually, that flow emptied the arteries. They became pale, indicating

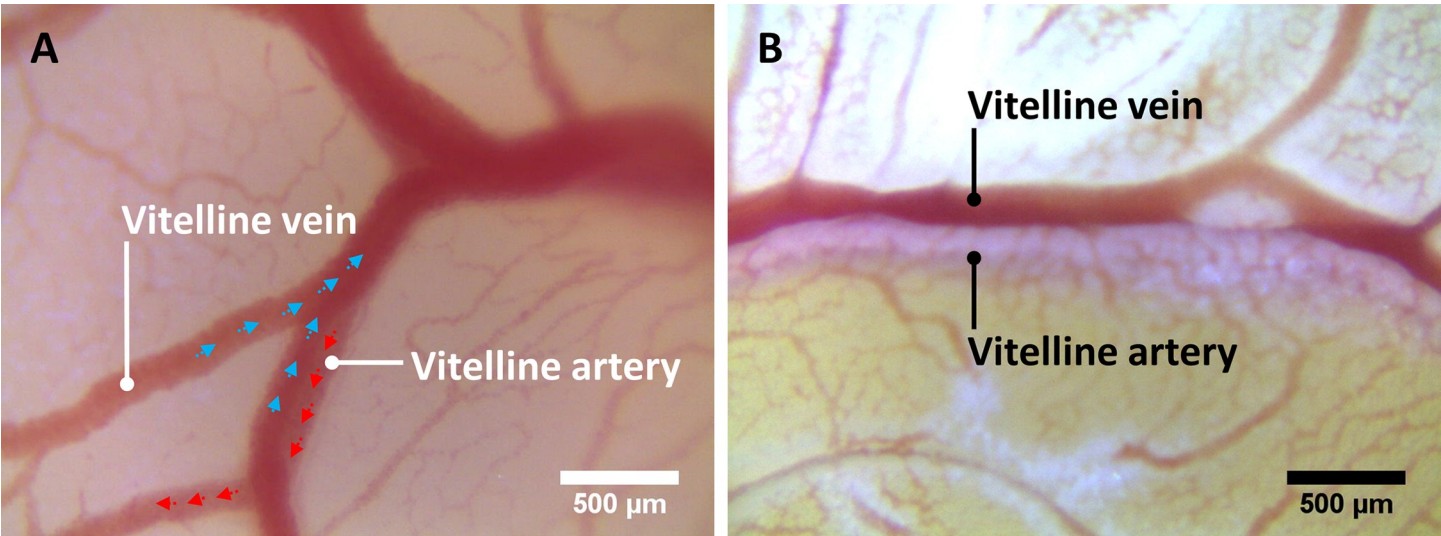

**Fig 3. Postmortem blood flow empties arteries.** (**A**) Vitelline artery and vein in live chick embryo. Artery and vein overlap, the artery lying partially beneath the vein. Blue arrows indicate the direction of venous flow, red arrows, arterial flow. (**B**) Vitelline artery and vein in a postmortem chick embryo, 20 minutes after the heart had stopped. Blood tends to exit arteries, leaving them pale.

loss of red-blood cells (Fig 3). Besides that decrease in red-cell density, arterial diameters diminished. Emptying of arteries occurred consistently, in all experiments.

Postmortem blood flow was not associated with vascular contraction. At the beginning stage of postmortem flow, we did observe moderate vascular contraction in the main artery leading from the heart; at later stages, however, it ceased, the width of the blood stream remaining constant. This implies that the arteries had stopped contracting (S5 Movie, 0:02:03–0:02:17); yet, flow persisted.

Associated with the postmortem blood flow, a zone devoid of red blood cells gradually appeared just inside the vessel wall. An example can be seen at the arterial wall (Fig 4; S5 Movie 0:01:00–0:01:13). Although the nature of this clear zone has not yet been explored, this cell-free zone may correspond to the so-called "exclusion zone" (EZ), previously reported [20,32–41].

### Effect of IR on postmortem and physiological blood flow

We next studied the effect of IR. During postmortem blood flow, velocity significantly increased during IR exposure, reaching a peak more than three times the control value after approximately three minutes. Following removal of the IR source, the velocity returned near to its original value (Fig 5). Thus, we could confirm the positive effect of IR on postmortem blood flow.

We also studied the effect of IR on blood flow in the physiological state, i.e., with the heart functioning. With IR application, venous flow velocity increased by ~30%, beginning almost immediately (Fig 6). After the IR source was removed, the venous flow diminished below the baseline. This positive effect of IR agrees with the results of previous studies [42,43].

In addition to the increased flow, however, there was also a change of heart rate, rising from ~100 to ~140 beats/minute. The question: does IR increase the venous blood flow by cranking up the heart rate? Or, by fueling a driving mechanism that potentially exists in the vessels? Conceivably, IR-induced increase of temperature could lead to a series of biochemical events that elevate the heart rate [44], which would in turn increase venous blood flow. On the other hand, if vessels can drive blood flow, then the direction of that flow should presumably point

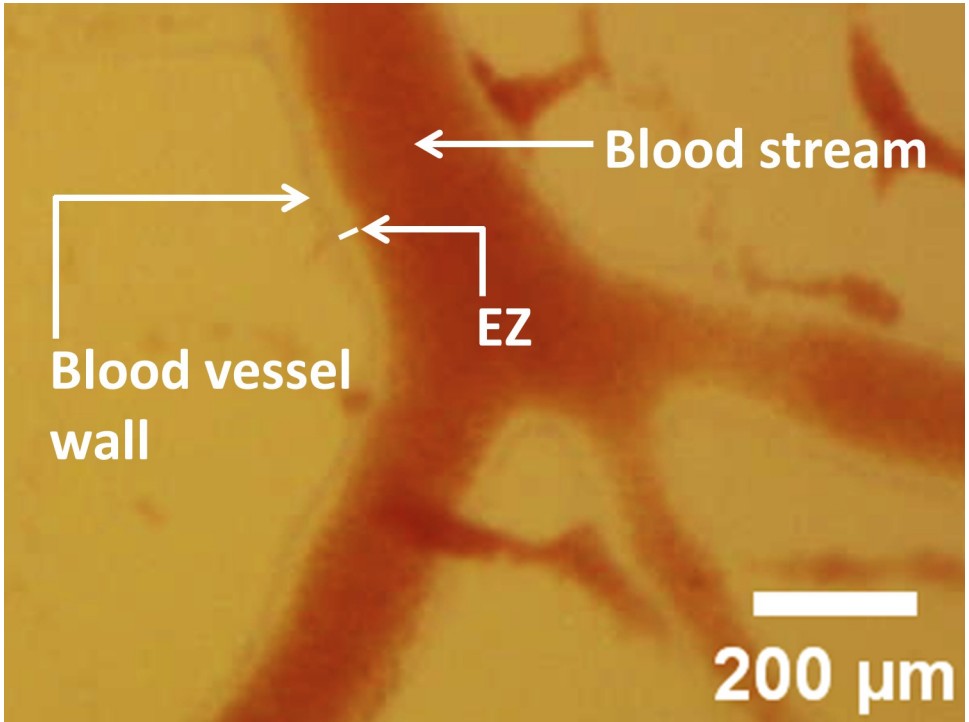

**Fig 4. Exclusion Zone (EZ) in vitelline arteries 150 seconds after cardiac arrest.** Note the erythrocyte-free region between the vessel wall and the boundary of the blood stream. The image is a snapshot from S5 Movie.

toward the heart. In that case, additional IR energy should enhance venous blood flow. Enhanced venous blood flow (venous return) can, in turn, affect the heart rate and contractility via mechanisms such as Bainbridge reflex or Frank-Starling mechanism [45,46]. Thus, the reciprocal causation between the cardiac output and venous return makes it difficult to distinguish whether the added IR impacted venous blood flow via the heart or via the blood vessels.

## Effect of IR deficiency on physiological blood flow

To further study how IR energy impacts the circulation, we investigated the effect of IR diminution. IR, above ambient, had been provided by the onstage incubator. To realize an IR deficiency, we removed the embryo from the warm onstage incubator, exposing it to the room-temperature laboratory environment.

At the lower temperature thereby achieved, the heart rate decreased appreciably, from ~100 to ~10 beats/minutes. Blood flow in the capillaries came to a complete halt, with no synchronous response to the heartbeat. Only in shunt vessels–larger blood vessels that directly connect arteries with veins–the blood flowed in a pulsed manner that corresponded to the heartbeat, providing an open passage for the circulation. This open passage indicates that the heart, albeit weak, still drives the blood flow. Yet, without sufficient IR, the heart alone is apparently insufficient to drive the blood flow through the capillaries.

Upon further observation, we found that under the diminished IR condition, the red blood cells were trapped in capillaries. Compared to the case with sufficient IR (Fig 7A), the arteries and veins contained less blood, as indicated by their lighter color and smaller diameter (Fig 7B). The capillaries, on the other hand, became darker in color, indicating more erythrocytes (Fig 7B). This observation implies that blood can enter the capillaries but cannot exit, i.e., the red cells get trapped in capillaries.

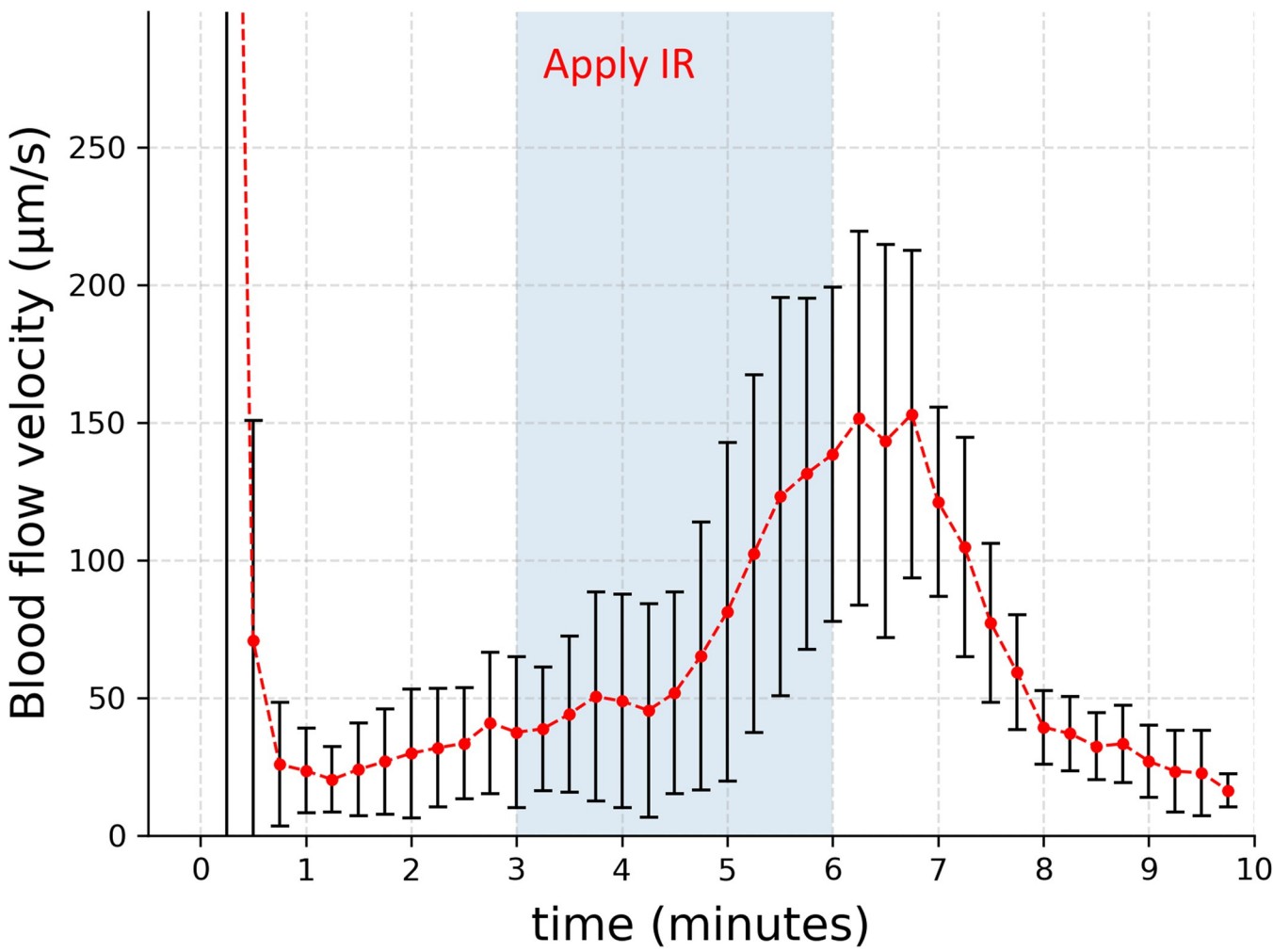

**Fig 5. Effect of IR exposure on venous postmortem blood flow ($n$ = 6).** Flow velocity increased appreciably after IR was turned on, diminishing when the IR source was removed. Error bars show standard deviations.

The effect of deficient IR was reversible. When the embryo was returned to the incubator, the capillaries, blocked by red blood cells, gradually opened, and blood flow resumed.

The accumulation of blood in the capillaries implies that vasomotor mechanisms, which are usually used to explain the impact of temperature on circulation [47], may not be the cause of the hindered circulation. Vasomotor mechanisms regulate blood flow by changing the diameter of the arterioles, thus controlling the amount of blood entering the capillary beds. At this developmental stage, however, the smooth muscles that control arteriolar diameter are not yet fully functional. If partially functional smooth muscles were to constrict the arterioles, then the blood should have difficulty entering, but not exiting the capillary beds.

In sum, without sufficient IR energy, blood failed to exit the capillaries. This implies a driving mechanism that uses IR energy present in capillaries—a conclusion in line with previous research [48].

## Discussion

Using the vitelline vascular network of the early-stage chick embryo as a model, we confirmed that blood continued to flow, following its natural course, even when the heart ceased

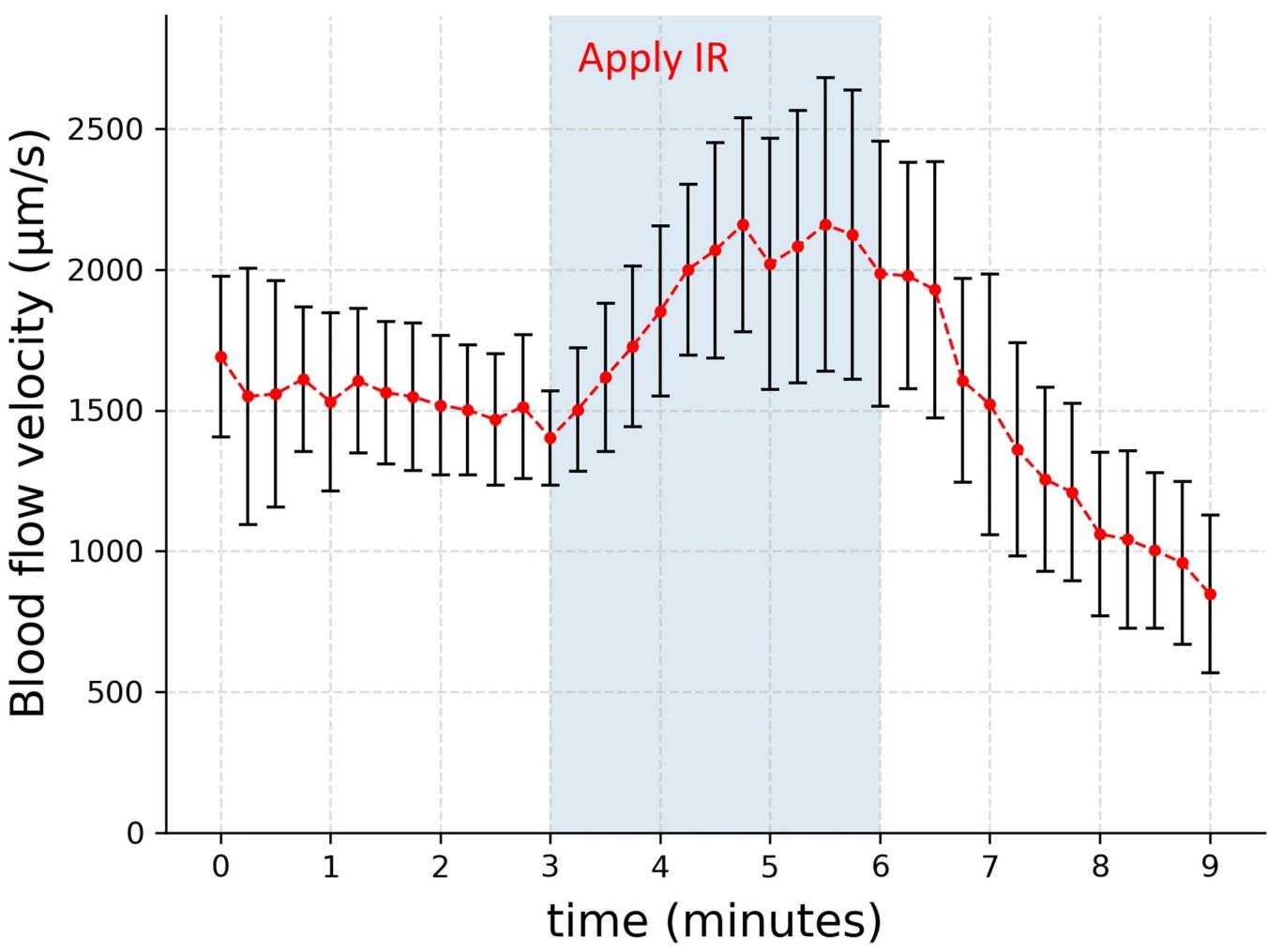

**Fig 6. Effect of IR on live chick-embryo venous blood flow.** Venous blood flow velocity increased appreciably when IR was turned on, and then decreased after removal ($n = 5$). Error bars show standard deviations.

pumping. IR energy appears to fuel this postmortem blood flow. We could also confirm the impact of IR energy on blood flow in the normal, physiological state.

Postmortem blood flow is a long-known phenomenon, previously reported in various animal models [4–7]. One feature of the postmortem flow is the tendency to deplete the arteries of blood, a finding confirmed here as well as in the older studies. Thus, when the Greeks found the arteries of the deceased devoid of blood, they considered the artery as an "air duct" (in Greek, "άρτηρία", in Latin characters, "*arteria*"). We now know that the artery is by no means a duct for air. Being a misnomer, the term "artery" may in fact have reflected the existence of postmortem blood flow.

## Trivial explanations?

Postmortem blood flow has been considered to be a consequence of one of two extraneous features: pressure originating from gravitational force; or, vascular contraction [6,13,21]. As a result of these potentially trivial explanations, the phenomenon has received little attention.

With the chick embryo as a model, we were able to exclude these two extraneous factors. As for the gravitational hypothesis, the model's architecture would seem incompatible. The

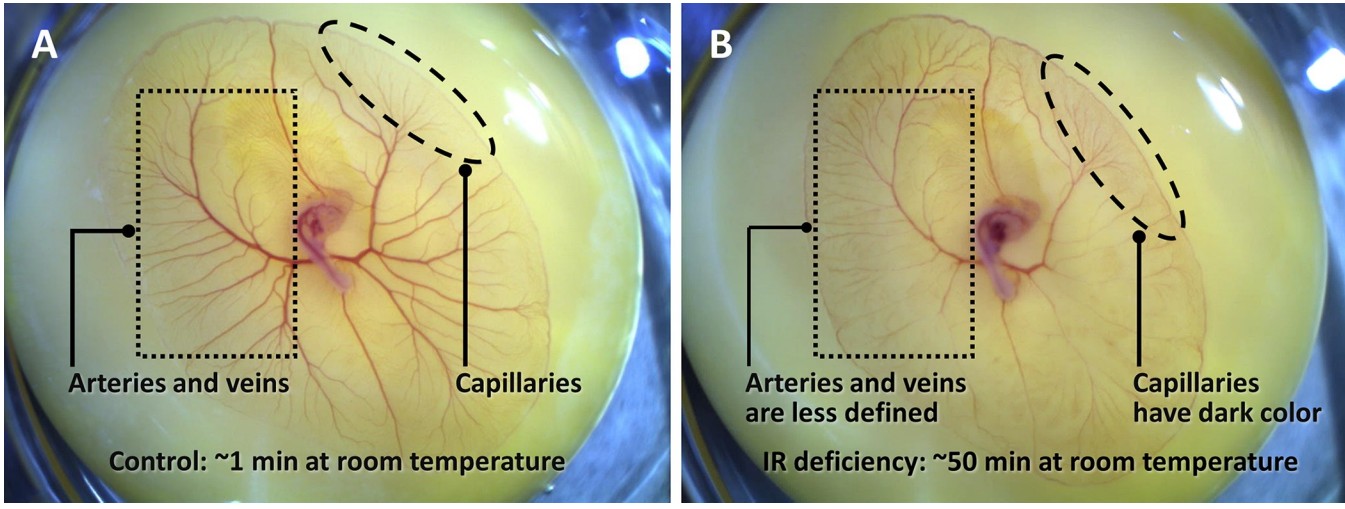

**Fig 7. Effect of IR deficiency on live chick embryo circulation.** (**A**) Vitelline vascular network with embryo just removed from onstage incubator. (**B**) Same as left but exposed to room temperature for ~50 minutes. Compared to Fig 7A, the arteries and veins (black dotted rectangle) appear thinner and less defined, indicating that those vessels contained less blood. Capillaries (black dashed oval), on the other hand, appear darker, indicating more blood.

embryonic vitelline vascular network lies essentially in a 2D plane floating atop the yolk, with the venous network lying immediately above the arterial network. Hence, reaching the veins requires the arterial blood to flow *against* gravity, rendering the gravitational hypothesis unlikely. As for the vascular contraction explanation: if the postmortem blood flow were caused by vascular contraction, then, that flow should be associated with the contracting of blood vessels. However, long after the modest vascular contraction had stopped, postmortem blood flow was observed to continue, with constant blood-stream width (see S5 Movie). Hence long-lasting flow was not coincident with the observed duration and degree of contraction. The two phenomena appear to be temporally distinct.

Thus, the potential artifacts that might underlie the presence of postmortem flow appear inadequate to account for the observation. Postmortem flow appears to be a genuine feature of the cardiovascular system, revealed only after the heart stops beating.

## Mechanism

In deducing the mechanism of postmortem flow, a critical feature is the role of IR energy. We found that IR enhanced postmortem flow. The energizing role of IR is a signature feature of the "surface-induced flow" phenomenon mentioned in the Introduction [20]. Hence, that phenomenon may well have relevance for propelling the flow.

Surface-induced flow is driven by chemical concentration gradients originating from physico-chemical reactions that occur at the tubular or vascular surface [18–20]. The vascular surface may release substances into the vessel, or remove them from the vessel, thereby altering the local intravascular concentration. Once an axial concentration gradient forms, flow is induced along that gradient [20]. Chemical potential energy is thus converted into kinetic energy.

We recently confirmed that intravascular gradients may form in two ways [20]: (i) material exchange across the vascular wall; or (ii) water-surface interaction. As for (i), solutes and solvents naturally enter/exit capillaries through the vascular walls; this is a fundamental feature of capillary function. Regarding water-surface interaction (ii), water interacts with many hydrophilic surfaces. The signature feature of such interaction is the formation of an 'exclusion zone'

(EZ), an interfacial region of water that excludes particles and solutes [33,35,49,50]. The EZ, often termed the "fourth phase" of wafer [50], is generally negatively charged, the charge achieved by releasing protons to the region beyond the EZ as it forms [33,35,49]. In tubes made from EZ-forming materials, the released protons accumulate in the core of the tube, creating an axial proton gradient, which can then drive flow [18–20]: This can occur via a 'solute-drags-solvent' model, as described by Kedem & Katchalsky [51]. External electromagnetic radiation fuels EZ formation, IR being the most effective wavelength [19,20,52]. For surface-induced flow of both kinds, we confirmed that IR has a positive effect on flow *in vitro* [20].

### Water-surface interaction in the vascular system?

The presence of EZ has been confirmed next to a wide range of materials including proteins and polysaccharides [20,33,38,40]. These are major components of blood vessels and their inner surface linings [53,54].

Indeed, in the physiological state, an annular "cell-free layer" just inside the vessel wall has long been known. This region is devoid of red blood cells [55,56]. Given the ability of blood-vessel components [53,54] to nucleate the growth of EZ [20], and the known ability of EZs to exclude particles, we hypothesize that the origin of that cell-free layer may lie in the presence of an exclusion zone just inside the vessel wall. We observed what appears to be an EZ immediately inside the vascular wall (Fig 4): red blood cells were excluded. Hence, evidence indicates that water-surface interaction exists in the vascular system.

Thus, postmortem blood flow may arise essentially from a surface-induced flow, driven by water-surface interaction. Material exchange, the companion mechanism, is unlikely to play a role in post-mortem capillaries, since the pressure required to drive that exchange [57] is largely absent post-mortem. Hence, the water-surface interaction remains as the more likely mechanism operating at the capillary level.

In the absence of any alternative post-mortem driving mechanism that we can identify, we argue that this water-surface interaction may well be the dominating driver of blood flow in situations in which the heart has stopped beating. In the physiological situation, when the heart continues to beat, the surface-water phenomenon may merely augment the heart-driven flow (Fig 6).

### Flow direction: Model considerations

The expected direction of any surface-induced flow would seem unclear *a priori*. If such a flow mechanism is to play a functional role in the vascular system, then it must propel flow in the natural direction, from arteries to veins.

To appreciate the direction expected in the vascular system, we first consider two representative examples that help identify the determining factors. We then examine those factors in the context of the entire vascular system, where we analyze the predicted direction and compare it with the observed direction.

Fig 8A shows the first of the two examples, involving a tube with uniform physico-chemical surface activity along its length, but with some taper. Suppose this surface activity results in substances entering the tube at the same rate per unit surface area throughout the length of the tube (green arrows). Compared to the wider region of the tube, the narrower region has a higher surface-to-volume ratio, and therefore, a higher concentration. Thus, a concentration gradient will form within the tube, pointing from the narrower to the wider region. The natural tendency for concentrations to even out should drive flow toward the wider region. Additional explanation can be found in the Supporting Information, in the section of S5 Appendix in S1 File.

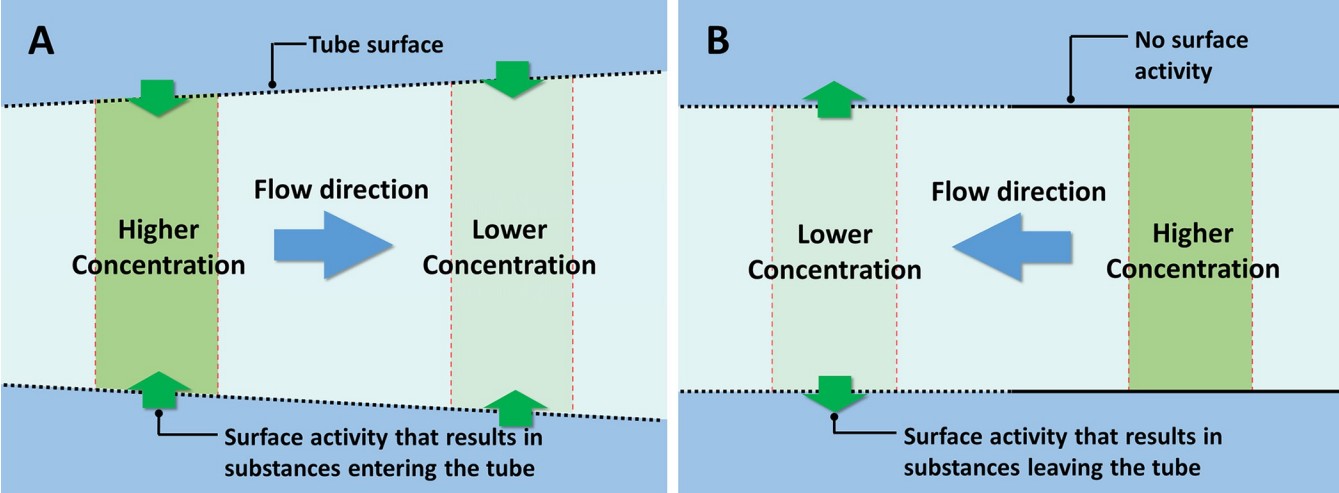

**Fig 8. Potential driving mechanism of surface-induced flow.** (**A**) In a tube with non-uniform diameter but uniform surface activity, an axial concentration gradient can form [20]. (**B**) An axial concentration gradient can also form in a tube with uniform diameter but nonuniform surface activity.

On the other hand, a surface activity that removes molecules from the tube instead of introducing them into the tube should drive flow in the opposite direction, i.e., towards the narrower region.

Fig 8B shows the second example, a uniform tube with nonuniform surface activity along its length. Suppose, for example, that a portion of the tubular surface can remove substances from the tube (dotted lines), the remainder not (solid lines). With any such removal, the intratubular region with surface activity will have a relatively lower concentration. Thus, an axial concentration gradient will form, driving flow to the region with lower concentration. If the surface activity admits solutes instead of eliminating them, then the flow direction would be reversed.

To summarize the two conclusions above, the direction of the surface-induced flow should be determined by the direction of intratubular concentration gradient, which is in turn determined by the following two factors: (i) *type of surface activity*—whether the tubular surface introduces or removes substances; and (ii) *tube geometry*—how the surface-to-volume ratio changes along the tube's length.

Knowing those determining factors, we may then consider the expected directions of surface-induced flows in the full vascular system—driven either by water-surface interaction, or, material exchange.

## Flow direction in the full vascular system

In considering blood-flow direction in the entire vascular system, we first note the results of previous research, which have shown that water-surface interaction commonly releases protons [33,35,49]. If the vascular wall operates similarly, then, according to the model of the previous section, the issue of flow direction should rest principally with the architecture of the network.

The vascular network has a hierarchical branching structure (Fig 9A), with each hierarchical level viewed as a set of vessels with similar diameter. The surface-to-volume ratio of these blood vessels is inversely proportional to their average diameter (for detailed analysis, see supplemental materials). This feature allows us to predict flow direction (assuming uniform behavior in all vessel walls). Based on the mechanism described in Fig 8A, blood should consistently flow from narrower to wider vessels.

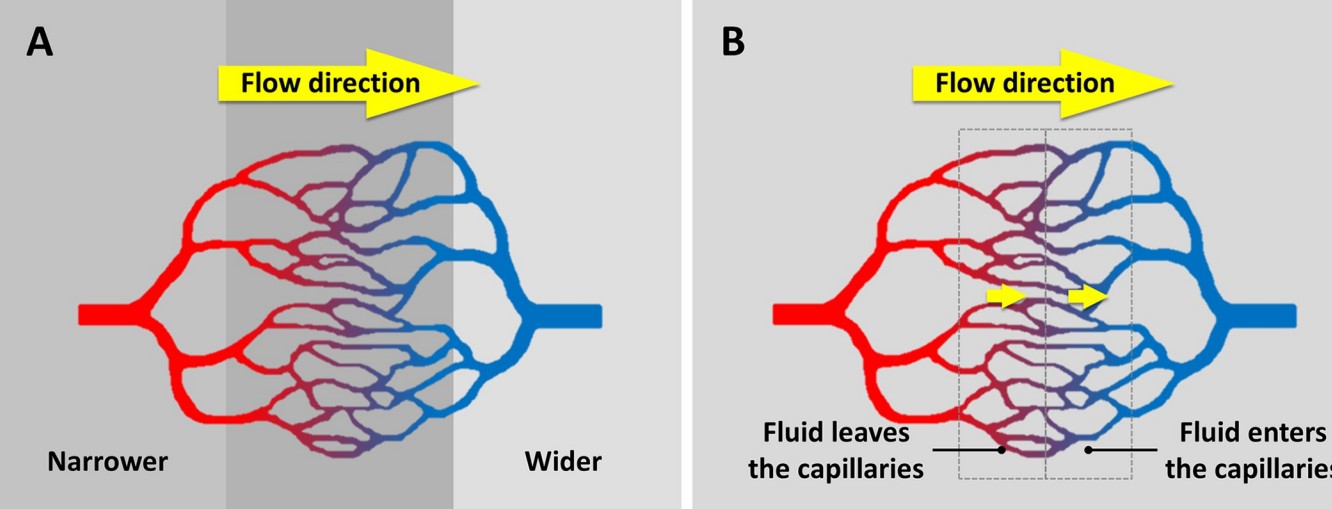

**Fig 9. Surface-induced flow in a capillary bed.** (**A**) Compared to venules (blue), the arterioles (red) are narrower; thus, the water-surface interaction drives capillary flow from arteries to veins. (**B**) As fluid (mainly water) leaves the capillaries through the capillary wall at the arteriolar end, a surface-induced flow is generated pointing from the arteries to the capillaries. Meanwhile, as fluid enters the capillaries at the venular end, a surface-induced flow is generated, pointing from the capillaries to the venules.

The same is predicted in capillaries (Fig 9A). Capillary flow should depend on the relative size of input and output vessels. Arterioles (red) have a narrower average diameter than venules (blue)–approximately one third of venule diameter [58]. For a capillary bed lying in between arterioles and venules, then, the blood should flow from narrower arterioles to wider venules.

The predicted blood flow direction should therefore follow its natural direction in capillaries. It should also do so in veins, where the natural progression is from narrower to wider. But predicted flow in arteries should be opposite to the natural, because size progressively decreases from the heart to the periphery. As the model predicted, the flow in the large, near-heart arteries was indeed opposite to the natural direction just after the heart stopped beating (S5 Movie, 0:00:51–0:01:35). Hence, model predictions appear to match experimental observations for all vessel beds.

If the flow in arteries is against the flow in the capillaries and the veins, a natural question is: who plays the dominating role? The answer should be the capillaries and the veins: compared to the arterioles, the venules are higher in number [58]; thus, more venules can generate flow. This conclusion is verified by the dynamics of the postmortem arterial blood flow. Postmortem flow in larger arteries was originally in the reversed direction, not the natural direction. Yet, the flow gradually resumed its natural direction from the peripheral region of the arterial network, indicating that the blood flowed into the capillaries and the veins. As the non-beating heart stopped replenishing blood to the arteries, ultimately, the arteries emptied. The emptied arteries indicate that the flow-driving capacity of capillaries and veins exceeds that of the arteries. Thus, all blood vessels drive the blood towards the natural direction.

Having considered water-surface interaction, we move to the other surface activity that could potentially drive blood flow: material exchange. That exchange occurs specifically in the capillaries. There, materials may exchange mainly in two forms: (i) *trans-vascular flow* ("bulk flow"), which refers to the fluid movement across the capillary wall; and (ii) *diffusion*, which refers to gas and electrolyte movement across the capillary wall [57].

Both types of exchange can generate an axial material gradient between the arterioles and venules; and that gradient can drive flow through the capillaries. Regarding (i): Fluids enter and

exit a capillary at different locations. At its arteriolar end, hydrostatic pressure drives fluid out through the capillary wall (Fig 9B, left dashed rectangle), while at the venular end, osmotic pressure draws fluid into the capillaries (Fig 9B, right dashed rectangle) [57]. These activities alter the "water-concentration potential (or, water potential)" in the capillaries. Driven by this potential, water should flow from arteries to capillaries (as described in Fig 8B), while a capillary-to-vein-directed flow should occur at the venular end (the reverse of that described in Fig 8B).

Regarding (ii), diffusion-driven capillary flow, consider the exchange of gas as an example. In the systemic circulation, as the arterial blood enters the capillaries, the oxygen carried by blood diffuses into the surrounding tissues. Meanwhile, the $CO_2$ generated from those tissues diffuses into the capillaries. Thus, an axial $O_2/CO_2$ gradient forms across the capillary bed. In the pulmonary circulation, an $O_2/CO_2$ concentration gradient will form between the arterioles and venules in a similar manner. Theoretical analysis has suggested that such concentration gradients may drive capillary flow towards the veins, by so-called macromolecular chemotaxis [59]. Thus, both trans-vascular flow and gas diffusion, two major forms of material exchange, may drive capillary flow from the arteries to the veins.

In sum, analysis of expected blood flow directions, driven by either water-surface interaction or material exchange, predict well-defined directionality. Both activities enable blood vessels to drive the blood in the natural direction.

## IR energy for facilitating venous return

In capillaries and veins, surface-induced flow drives blood towards the heart, facilitating venous return. A non-trivial amount of energy is needed: with capillary diameters sometimes narrower than a red blood cell, the resistance within each capillary is high. With this high resistance, red blood cells may stall, or even flow in a reverse direction [60]; yet when blood returns to the heart, it must flow at the same overall rate as the blood that leaves the heart.

To facilitate that return, IR energy can be used, via the surface-induced flow mechanism. Surface-induced flow converts chemical potential energy into kinetic energy. In this process, IR can operate two ways. It can: (i) accelerate proton production by enhancing water-surface interaction [52]; and, (ii) accelerate the conversion of chemical potential energy to kinetic energy by elevating the temperature. In this context, each capillary may be viewed as a flow generator that runs on IR energy [20], sending the blood back to the heart.

IR energy is one of the most available energy sources. It can be both endogenous and exogenous: In terms of endogenous sources in the body, metabolic activities of living organisms generate heat, which according to Planck's law, can be emitted as IR energy [43,61]. In terms of exogenous sources, approximately 50% of solar energy received by earth is in the form of IR [62]. Thus, capillaries have access to abundant IR energy for driving the flow, despite their high resistance.

We employ the term "IR" over the term "heating" for the following two reasons. First, the term IR contains wavelength information. This information is pertinent because the effect of light on surface-water interaction is wavelength-dependent, the IR range being the most effective [52]. Heating, on the other hand, does not contain wavelength information. Second, our focus is on the *energy* for driving blood circulation. "IR" represents a form of energy, while heating (or temperature) emphasizes the consequence of absorbing energy. For those reasons, we chose to use the term "IR."

## Both the heart and the blood vessels drive the circulation

The importance of the vessels' contribution to driving blood flow is shown by the effect of IR deficiency. In the intact chick-embryo model, where neither any muscle pump nor respiratory

pump is present to assist venous return, blood failed to flow out of the capillaries when IR energy was deficient (Fig 7B). Thus, IR energy, which can be used by the blood vessels, apparently plays a critical role in maintaining the normal circulatory function. The results show that When IR is deficient, the heart alone cannot do the job.

Thus, as the two essential components of the circulatory system, the heart and the blood vessels, evidently work together to drive the blood flow. The heart pumps blood into the vessels; the vessels amplify the flow, returning the blood back to the heart.

## Pathophysiological implications

Light enhances biological activity. Visible light (385–750 nm) and monochromatic infrared energy (750–1300 nm) are reported to have diverse health benefits, including wound healing, boosting of blood circulation, and reversal of peripheral sensory neuropathy [63–68]. Far infrared can increase skin microcirculation [69] and improve peritoneal membrane function for dialysis patients [70].

Light's role in overall circulatory function seems particularly important. Enhanced flow derived from the vessel-driving mechanism can reduce the load on the heart, potentially allowing a damaged heart to heal. This effect is confirmed: A far-infrared sauna, whose application should build EZ water, improves the prognosis in chronic heart failure [71].

## Perspectives

The heart cannot be the sole driver of blood flow. If it were, then stopping the heart would stop all flow. Persisting flow after the heart stops beating has been reported in multiple studies [4–10]. We found the same. In our case, various potential artifacts could be ruled out. Thus, some mechanisms must supplement the cardiac driver.

Here, we demonstrated the existence of that second driving mechanism: surface-induced flow, originating in the blood vessels themselves. The circulation appears to be driven not only by the heart, but also by the vessels themselves. The fuel for this second driving mechanism lies in infrared energy, which emerges naturally from metabolic heat, as well as from external sources.

For any organism, the most fundamental requirement is metabolism. Metabolic activity generates heat. The heat is released in the form of IR energy, which can drive flow and hence nourish the tissues. Thus, metabolism facilitates the circulation, while the circulation facilitates metabolism. The two phenomena enable one another.

Appreciating the existence of this second circulatory driver opens the door to fresh understanding of cardiovascular disease, as well as to unforeseen therapies for combatting that disease. Thus, we may anticipate novel therapies appearing in the future.

## Supporting information

**S1 File. Contains the appendices and supporting figures.**
(DOCX)

**S1 Movie. Postmortem venous blood flow at ROI.**
(MOV)

**S2 Movie. Postmortem venous blood flow at ROI, with brief reversal.**
(MOV)

**S3 Movie. Postmortem venous blood flow at the boundary of the vascular network.**
(MOV)

**S4 Movie. Postmortem blood flow in capillaries.**
(MOV)

**S5 Movie. Postmortem arterial blood flow dynamics.** This footage was taken on an embryo with the vascular network deformed, so as to expose the vitelline artery.
(MP4)

## Acknowledgments

We thank, H. Lai, R. J. Wilkes, W. Kaminsky, K. Böhringer and Q. Yu for discussions on the experimental design; R. Wang and V. Kalchenko for discussions on model selection; D. Dabiri and W.H. Tien for discussion on data quantification; H. Liu, J. Huang, A. Traynor-Kaplan, M. Kowacz, A. Wang, T. Ye, L. Colton, S. Landefeld, A. Pinhas, J. Wu, A. Sharma, R. Sharma, Z. Wan and Y. Zhao for comments on the manuscript; H. Lin and X. Xin for suggestions on statistical analysis; R. Hua for help on the figures and continuous support to the first author to finish his PhD.

## Author Contributions

**Conceptualization:** Zheng Li, Gerald H. Pollack.

**Data curation:** Zheng Li.

**Formal analysis:** Zheng Li.

**Funding acquisition:** Gerald H. Pollack.

**Investigation:** Zheng Li.

**Methodology:** Zheng Li.

**Project administration:** Zheng Li.

**Resources:** Zheng Li, Gerald H. Pollack.

**Software:** Zheng Li.

**Supervision:** Zheng Li, Gerald H. Pollack.

**Validation:** Zheng Li.

**Visualization:** Zheng Li.

**Writing – original draft:** Zheng Li, Gerald H. Pollack.

**Writing – review & editing:** Zheng Li, Gerald H. Pollack.

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
