## [Decision Letter · Decision Letter 0]

8 Feb 2023

PONE-D-22-35747On the Driver of Blood Circulation Beyond the HeartPLOS ONE

Dear Dr. Li,

Thank you for submitting your manuscript to PLOS ONE. After careful consideration, we feel that it has merit but does not fully meet PLOS ONE’s publication criteria as it currently stands. Therefore, we invite you to submit a revised version of the manuscript that addresses the points raised during the review process.

We look forward to receiving your revised manuscript.

Kind regards,

Peter R. Corridon

Academic Editor

PLOS ONE

Journal Requirements:

2. Thank you for stating in your financial disclosure:  

"This work was supported by an NIH Transformative grant 5R01GM093842, received by GHP. https://www.nih.gov/

This work was also supported by private support, received by GHP. No websites available.

PLOS ONE requires you to include in your manuscript further information about the funder so that any relevant competing interests can be assessed. Please respond to the following questions:

a) Please state whether any of the research costs or authors' salaries were funded, in whole or in part, by a tobacco company (our policy on tobacco funding is at http://journals.plos.org/plosone/s/disclosure-of-funding-sources)  

b) Please state whether the donor has any competing interests in relation to this work (see http://journals.plos.org/plosone/s/competing-interests) . 

c) Please state whether the identity of the donor might be considered relevant to editors or reviewers’ assessment of the validity of the work.

d) If the donors have no perceived or actual competing interests, please state: “The authors are not aware of any competing interests”. 

This information should be included in your cover letter. We will amend your financial disclosure and competing interests on your behalf.

Reviewers' comments:

Reviewer's Responses to Questions

**Comments to the Author**

1. Is the manuscript technically sound, and do the data support the conclusions?

Reviewer #1: Yes

Reviewer #2: Yes

2. Has the statistical analysis been performed appropriately and rigorously? 

Reviewer #1: No

Reviewer #2: Yes

3. Have the authors made all data underlying the findings in their manuscript fully available?

Reviewer #1: No

Reviewer #2: Yes

4. Is the manuscript presented in an intelligible fashion and written in standard English?

Reviewer #1: Yes

Reviewer #2: Yes

5. Review Comments to the Author

Reviewer #1: See attached file.

The paper reports evidences of the existence of a flow-driven mechanism using infrared energy. Experiments are performed using a three-day-old chick-embryo model. The authors demonstrate that when infrared is applied the blood flow increases and when infrared energy is decreases the flow also decrease. Furthermore, when the heart of the chicken embryo is stopped the blood keeps flowing when infrared energy is applied.The experiments are nicely performed and the results reported are relevant. However the models proposed are mostly speculative and do not contribute to revel the mechanism leading to the blood flow when infrared energy is applied. In my opinion, a more elaborate mechanism should be proposed to be able to publish the manuscript in PLOS.\\\\

Some comments are given below,

\\begin{enumerate}

\\item When the embryo heart was stopped, IR energy was able to fuel the blood flow. In the experiments reported, IR was applied for a few minutes. I wonder how long the IR energy is able to generate a blood flow after the heart stops.

\\item The authors claim that surface-induced flow is the mechanism, triggered by IR energy, that explain the postmortem blood flow. Two kinds of surface-induced flows are proposed: i) material exchange across the vascular walls and ii) water-surface interaction. The explanations of both mechanisms are very speculative and more precise and quantitative description should be provided. To say that "Chemical potential energy is converted in kinetic energy" is ambiguous, can the chemical potential energy be estimated? how is the energy converted in kinetic energy?

\\item I am not sure that the mechanism described in figure 4A, in a tube with uniform physico-chemical surface activity, is correct. If we applied the mass conservation equation to the narrow and the wide section we obtain that the concentration is the same in both sections, since the flowrate is the same. In fact, applying mass conservation to the narrow section,

\\[m_{in}+m_s= m_{on}\\] where $m_{in}$ is the mass flowrate at the input, $m_s$ the mass flowrate of the substance entering the control volume and $m_{on}$ the mass flowrate exiting the narrow control volume. The substance concentration is therefore $C_n= m_s/(m_{in}+m_s)$. Performing the same analysis to the wider section

\\[m_{iw}+m_s= m_{ow}\\] where $m_{iw}$ is the mass flowrate at the input of the wider section, $m_s$ the mass flowrate of the substance entering the control volume and $m_{ow}$ the mass flowrate exiting the wide control volume. The substance concentration is therefore $C_w= m_s/(m_{iw}+m_s)$. Since $m_{in}=m_{iw}$ we should get that $C_n=C_w$

\\end{enumerate}

In general, although I consider that the topic is of scientific importance, the mechanisms leading to the blood flow should be discussed in more detail, providing quantitative information, before the manuscript can be accepted.

\\end{document}

Reviewer #2: In my opinion, the topic is of interest. However, the current version still needs improvement.

Besides, in order to expedite the processing of the revised manuscript, the authors had better provide a detailed point-by-point response letter and all the changes have been highlighted in some color in the new revised version. Finally, I suggest a minor revision and my comments are as follows:

1. In the abstract section, the quantitative findings of the study can also be briefly mentioned.

2. Give the title and sub-headings in order.

3. Include references to mathematical expressions that are not derived from this work.

4. Correct the typos in the entire text and improve the English level of the article.

5. Organize references in accordance with journal standards.

6- Consider the following references;

o Thermal radiation and surface roughness effects on the thermo-magneto-hydrodynamic stability of alumina-copper oxide hybrid nanofluids utilizing the generalized Buongiorno's nanofluid model, J. Therm. Anal. Calorim. 143 (2021) 1201-1220. doi:10.1007/s10973-020-09488-z.

o Meta-analysis on thermo-migration of tiny/nano-sized particles in the motion of various fluids, Chinese J. Phys. 68 (2020) 293-307. doi:10.1016/j.cjph.2019.12.002.

o Numerical spectral examination of EMHD mixed convective flow of second-grade nanofluid towards a vertical Riga plate using an advanced version of the revised Buongiorno's nanofluid model, J. Therm. Anal. Calorim. 143 (2021) 2379-2393. 21 doi:10.1007/s10973-020-09865-8.

o New insights into the dynamics of alumina-(60% ethylene glycol + 40% water) over an isothermal stretching sheet using a renovated Buongiorno's approach: A numerical GDQLLM analysis, Int. Commun. Heat Mass Transf. 133 (2022) 105937. doi:10.1016/j.icheatmasstransfer.2022.105937.

o Hydrothermal and mass impacts of azimuthal and transverse components of Lorentz forces on reacting Von Kármán nanofluid flows considering zero mass flux and convective heating conditions, Waves in Random and Complex Media. (2022) 1-22. 22 doi:10.1080/17455030.2022.2136413.

o Dynamics of radiative-reactive Walters-B fluid due to mixed convection conveying gyrotactic microorganisms, tiny particles experience haphazard motion, thermo-migration, and Lorentz force, Phys. Scr. 96 (2021) 125239. doi:10.1088/1402-4896/ac2b4b.

o Importance of Exponentially Falling Variability in Heat Generation on Chemically Reactive Von Kármán Nanofluid Flows Subjected to a Radial Magnetic Field and Controlled Locally by Zero Mass Flux and Convective Heating Conditions: A Differential Quadrature, Front. Phys. 10:988275 (2022) 1-17. doi:10.3389/fphy.2022.988275.

o A Brief Technical Note on the Onset of Convection in a Horizontal Nanofluid Layer of Finite Depth via Wakif-Galerkin Weighted Residuals Technique (WGWRT), Defect Diffus. Forum. 409 (2021) 90-94.

6. PLOS authors have the option to publish the peer review history of their article (what does this mean?). If published, this will include your full peer review and any attached files.

Reviewer #1: No

Reviewer #2: No

---

## [Author Response · Author response to Decision Letter 0]

12 May 2023

Response to the Decision Letter

On Manuscript Named

“On the Driver of Blood Circulation Beyond the Heart”

Zheng Li, Ph.D. and Gerald Pollack, Ph.D.

Department of Bioengineering, University of Washington, Seattle, Washington, USA.

*Correspondence to: zhl@u.washington.edu, ghp@u.washington.edu

We thank the editor and referees for their careful review of our manuscript. We have now fully responded to the reviewers’ comments. Our responses are presented below. 

Response to Reviewer #1

The paper reports evidences of the existence of a flow-driven mechanism using infrared energy. Experiments are performed using a three-day-old chick-embryo model. The authors demonstrate that when infrared is applied the blood flow increases and when infrared energy is decreases the flow also decrease. Furthermore, when the heart of the chicken embryo is stopped the blood keeps owing when infrared energy is applied. The experiments are nicely performed and the results reported are relevant. However the models proposed are mostly speculative and do not contribute to revel the mechanism leading to the blood flow when infrared energy is applied. In my opinion, a more elaborate mechanism should be proposed to be able to publish the manuscript in PLOS.

1. When the embryo heart was stopped, IR energy was able to fuel the blood flow. In the experiments reported, IR was applied for a few minutes. I wonder how long the IR energy is able to generate a blood flow after the heart stops.

Before answering this question, we would like to clarify the IR energy source(s) during the postmortem blood flow study. IR energy came from two sources: 1) The on-stage incubator, which has a temperature of 37 C° and was kept on during the entire experiment. This provided base-line IR energy. 2) The infrared lamp. This lamp was only turned on for a few minutes during the experiment, and provided additional IR energy, aimed to demonstrate that IR energy could enhance the postmortem blood flow. 

Back to the question regarding how long the IR energy can maintain the flow after the heart stops: For our experimental setup, with the IR energy coming from the on-stage incubator, we observed a postmortem blood flow lasting for approximately 50 minutes. 

The duration may have been artificially shortened: the blood vessels drive the postmortem blood flow through the veins and towards the heart. But the stopped heart became an obstacle to flow, forcing the blood to accumulate in the veins, and thereby cutting off flow. Thus, it is possible that the blood vessels could transport the blood for more than just the 50 minutes reported; however, the duration may have been artificially shortened by the resistance of the non-beating heart.

--

2. The authors claim that surface-induced flow is the mechanism, triggered by IR energy, that explain the postmortem blood flow. Two kinds of surface-induced flows are proposed: i) material exchange across the vascular walls and ii) water-surface interaction. The explanations of both mechanisms are very speculative and more precise and quantitative description should be provided. To say that "Chemical potential energy is converted in kinetic energy" is ambiguous, can the chemical potential energy be estimated? how is the energy converted in kinetic energy?

We appreciate that the reviewer brings up the question of quantitative description of the flow-driving mechanism. We had been pondering that question for some time. The calculation, unfortunately, involves multiple assumptions that remain uncertain; hence we are hesitant to draw any firm quantitative conclusions. That being said, we present preliminary thoughts on the lowest and highest estimates:

Lowest estimate: While the heart was still beating, the blood-flow velocity in anterior/posterior vitelline vein was ~1,500 μm/s – the physiological value. After the heart stopped beating, it dropped to ~25 μm/s. Thus, blood vessels are capable of driving at least 1.7% of the physiological blood flow. 

After the heart stopped beating, however, the non-pumping heart posed a flow obstacle. It restricts flow. Thus, the aforementioned 1.7% can be taken as a conservative estimate. It is worth noting that the material-exchange-driven flow mechanism needs pressure to operate. When the heart stops beating and there is no longer any ventricular pressure, this mechanism can no longer operate. While we could not quantify the magnitude of this flow component, we are confident to say that the sum of the material-exchange-driven flow and the surface-water-interaction-driven flow amounts to more than 1.7% of the total flow.

Highest estimate: Red blood cells must be driven through capillaries that are sometimes narrower than the RBCs that need to pass through. Therefore, those cells must get squeezed during their passage. Videos confirm erythrocyte bending in capillaries. Considering the likely energy required to bend each blood cell, the overall energy requirement could well exceed the capability of the heart (power = ~1.5 W). If so, then the required energy would therefore need to come from elsewhere.

Calculation of that energy requires multiple assumptions, which means that definitive conclusions on the contribution of the vascular mechanism may be difficult to draw. Nevertheless, the possibility exists that the vessels may contribute a very substantial fraction of the total driving energy – potentially providing virtually all the energy needed for venous return. To estimate just how much, additional experiments of different nature will be required.

In sum, like the reviewer, we are eager to determine just how much the vessel-driving system may contribute, but at present, too many unknowns exist to allow us to draw meaningful conclusions. It could vary between 1.7% of total and a very major percentage. Any attempt to estimate the amount without additional rigorous experimental study would be highly speculative, and potentially misleading. 

Regarding the issue of how the chemical potential energy can be converted to kinetic energy: From a physical-chemical standpoint, there are multiple ways for the chemical potential energy to be converted into kinetic energy. When a chemical gradient exists, for example, motion can be created by diffusion. When said particles exist in a chemical gradient field, diffusionphoresis is a known mechanism that can confer kinetic energy to the particles. In general, the key for chemical potential energy to be converted into kinetic energy is the presence of some kind of gradient. When blood vessels are transporting a water-based suspension and exchanging water and materials with its surrounding environment, such gradients can easily form.

Finally, we agree with the reviewer that a more quantitative description is needed, but more suitable for future studies. The scope of the current work is merely to demonstrate that blood vessels can use IR energy to transport blood, and to point to a plausible mechanism. Subsequent work, we hope, will allow us to draw quantitative conclusions.

3. I am not sure that the mechanism described in figure 4A, in a tube with uniform physico-chemical surface activity, is correct. If we applied the mass conservation equation to the narrow and the wide section we obtain that the concentration is the same in both sections, since the flowrate is the same. In fact, applying mass conservation to the narrow section, 

m_in+ m_s= m_on

where m_in is the mass flowrate at the input, m_s the mass flowrate of the substance entering the control volume and m_on the mass flowrate exiting the narrow control volume. The substance concentration is therefore C_n=m_s/(m_in+m_s) . Performing the same analysis to the wider section

m_iw+ m_s= m_ow

where m_iw is the mass flowrate at the input of the wider section, m_s the mass flowrate of the substance entering the control volume and m_ow the mass flowrate exiting the wide control volume. The substance concentration is therefore C_w=m_s/(m_iw+m_s). Since m_in= m_iw we should get that C_n= C_w.

In general, although I consider that the topic is of scientific importance, the mechanisms leading to the blood flow should be discussed in more detail, providing quantitative information, before the manuscript can be accepted.

In the reviewer’s argument, an important assumption is mass conservation, e.g., the mass flow at the inlet equals the mass flow at the outlet (m_in= m_iw). This is a valid assumption in the case of ordinary pressure- driven flow, and should be valid in larger blood vessels like arteries or veins. But when considering a capillary, flow happens across the tubular wall. Because substances regularly enter/leave the capillaries through the vascular wall, the mass flow of the inlet does not necessarily equal the mass flow at the outlet.

An extreme example is edema, where surrounding tissue draws water from the blood through the capillary wall. In this case, m_in cannot be equal to m_iw since much of the blood will leave the capillary through the wall.

We absolutely agree with the reviewer that quantification is desirable in future studies but feel hesitant to proceed at this stage because of the kinds of uncertainties outlined above, as well as those given in response to your earlier comment.

---

Response to Reviewer #2

Reviewer #2: In my opinion, the topic is of interest. However, the current version still needs improvement.

Besides, in order to expedite the processing of the revised manuscript, the authors had better provide a detailed point-by-point response letter and all the changes have been highlighted in some color in the new revised version. Finally, I suggest a minor revision and my comments are as follows:

1. In the abstract section, the quantitative findings of the study can also be briefly mentioned.

 We revised the abstract and reported our quantitative findings of the study. 

2. Give the title and sub-headings in order.

 We assigned proper levels to the title, headings, and sub-headings.

3. Include references to mathematical expressions that are not derived from this work.

 We only have mathematical expressions in “S2 Appendix. A calculation of surface-area-to-volume ratio of a single blood vessel and a vascular network.” All the math in that section is work of our own.

4. Correct the typos in the entire text and improve the English level of the article.

 We made corrections throughout the manuscript.

5. Organize references in accordance with journal standards.

 As suggested by the reviewer, we have reformatted our manuscript according to the journal’s standards, including the references.

6- Consider the following references;

o Thermal radiation and surface roughness effects on the thermo-magneto-hydrodynamic stability of alumina-copper oxide hybrid nanofluids utilizing the generalized Buongiorno's nanofluid model, J. Therm. Anal. Calorim. 143 (2021) 1201-1220. doi:10.1007/s10973-020-09488-z.

o Meta-analysis on thermo-migration of tiny/nano-sized particles in the motion of various fluids, Chinese J. Phys. 68 (2020) 293-307. doi:10.1016/j.cjph.2019.12.002.

o Numerical spectral examination of EMHD mixed convective flow of second-grade nanofluid towards a vertical Riga plate using an advanced version of the revised Buongiorno's nanofluid model, J. Therm. Anal. Calorim. 143 (2021) 2379-2393. 21 doi:10.1007/s10973-020-09865-8.

o New insights into the dynamics of alumina-(60% ethylene glycol + 40% water) over an isothermal stretching sheet using a renovated Buongiorno's approach: A numerical GDQLLM analysis, Int. Commun. Heat Mass Transf. 133 (2022) 105937. doi:10.1016/j.icheatmasstransfer.2022.105937.

o Hydrothermal and mass impacts of azimuthal and transverse components of Lorentz forces on reacting Von Kármán nanofluid flows considering zero mass flux and convective heating conditions, Waves in Random and Complex Media. (2022) 1-22. 22 doi:10.1080/17455030.2022.2136413.

o Dynamics of radiative-reactive Walters-B fluid due to mixed convection conveying gyrotactic microorganisms, tiny particles experience haphazard motion, thermo-migration, and Lorentz force, Phys. Scr. 96 (2021) 125239. doi:10.1088/1402-4896/ac2b4b.

o Importance of Exponentially Falling Variability in Heat Generation on Chemically Reactive Von Kármán Nanofluid Flows Subjected to a Radial Magnetic Field and Controlled Locally by Zero Mass Flux and Convective Heating Conditions: A Differential Quadrature, Front. Phys. 10:988275 (2022) 1-17. doi:10.3389/fphy.2022.988275.

o A Brief Technical Note on the Onset of Convection in a Horizontal Nanofluid Layer of Finite Depth via Wakif-Galerkin Weighted Residuals Technique (WGWRT), Defect Diffus. Forum. 409 (2021) 90-94.

Thank you. We have considered the references carefully and have inserted those references in the “Supporting Information” section.

---

## [Decision Letter · Decision Letter 1]

19 Jun 2023

PONE-D-22-35747R1On the driver of blood circulation beyond the heartPLOS ONE

Dear Dr. Li,

Thank you for submitting your manuscript to PLOS ONE. After careful consideration, we feel that it has merit but does not fully meet PLOS ONE’s publication criteria as it currently stands. Therefore, we invite you to submit a revised version of the manuscript that addresses the points raised during the review process.

We look forward to receiving your revised manuscript.

Kind regards,

Peter R. Corridon

Academic Editor

PLOS ONE

Journal Requirements:

Reviewers' comments:

Reviewer's Responses to Questions

**Comments to the Author**

1. If the authors have adequately addressed your comments raised in a previous round of review and you feel that this manuscript is now acceptable for publication, you may indicate that here to bypass the “Comments to the Author” section, enter your conflict of interest statement in the “Confidential to Editor” section, and submit your "Accept" recommendation.

Reviewer #1: (No Response)

Reviewer #2: All comments have been addressed

2. Is the manuscript technically sound, and do the data support the conclusions?

Reviewer #1: Yes

Reviewer #2: Yes

3. Has the statistical analysis been performed appropriately and rigorously? 

Reviewer #1: Yes

Reviewer #2: Yes

4. Have the authors made all data underlying the findings in their manuscript fully available?

Reviewer #1: (No Response)

Reviewer #2: Yes

5. Is the manuscript presented in an intelligible fashion and written in standard English?

Reviewer #1: Yes

Reviewer #2: Yes

6. Review Comments to the Author

Reviewer #1: The manuscript has been revised and the authors have responded to my comments. I understand that "the scope of the present work is merely to demonstrate that blood vessels can use IR energy to transport blood, and to point to a plausible mechanism". However, I regret that the authors are reluctant to provide a more quantitative information of he mechanisms leading to the motion. A characteristic velocity induced by the gradients of concentration can be established based on the characteristic length scale and the diffusivity. Nevertheless, I understand the difficulties alleged by the authors and I would agree to accept the publication of the paper. I only have a minor comment that should not be difficult to clarify in the revised version.

Minor comments,

- I am not sure to agree (or fully understand) the mechanisms discussed in the "Flow Direction: Model Considerations" section (Figure 4). I agree that the narrow section has a higher surface-to-volume ratio but I am not sure to understand that the flow should go from the narrow to the wide section. I would encourage the authors to include the analysis supported by the application of the mass conservation equation to control volumes in the narrow an the wide sections. In fact, if it is assumed that the surface activity results in substances entering the tube at the same rate per unit surface area throughout the length of the tube, a lower quantity of substance would enter the narrow section since the exchange area, $\\pi D L$, is smaller than in the wide section. Please clarify the discussion in this section supporting the information with the governing equations.

In general, although I consider that the topic is of scientific importance, some of the mechanisms leading to the blood flow are not sufficiently clear and should be clarified before the accepting the manuscript.

(See file attached)

Reviewer #2: The revision is OK because all comments are adressed.So, the paper can be accepted in its current form.

7. PLOS authors have the option to publish the peer review history of their article (what does this mean?). If published, this will include your full peer review and any attached files.

Reviewer #1: No

Reviewer #2: **Yes: **Abderrahim Wakif

---

## [Author Response · Author response to Decision Letter 1]

20 Jul 2023

Response to Remaining Issues in Revision Round 2

“On the Driver of Blood Circulation Beyond the Heart”

Zheng Li, Ph.D. and Gerald Pollack, Ph.D.

Department of Bioengineering, University of Washington, Seattle, Washington, USA.

*Correspondence to: zhl@u.washington.edu, ghp@u.washington.edu

We thank the editor and referees for their careful review of our manuscript. We have now fully responded to the reviewers’ comments. Our responses are presented below. 

Response to Reviewer #1

The manuscript has been revised and the authors have responded to my comments. I understand that ”the scope of the present work is merely to demonstrate that blood vessels can use IR energy to transport blood, and to point to a plausible mechanism”. However, I regret that the authors are reluctant to provide a more quantitative information of he mechanisms leading to the motion. A characteristic velocity induced by the gradients of concentration can be established based on the characteristic length scale and the diffusivity. Nevertheless, I understand the difficulties alleged by the authors and I would agree to accept the publication of the paper. I only have a minor comment that should not be difficult to clarify in the revised version. 

Minor comments, 

1. I am not sure to agree (or fully understand) the mechanisms discussed in the ”Flow Direction: Model Considerations” section (Figure 4). I agree that the narrow section has a higher surfaceto-volume ratio but I am not sure to understand that the flow should go from the narrow to the wide section. I would encourage the authors to include the analysis supported by the application of the mass conservation equation to control volumes in the narrow an the wide sections. In fact, if it is assumed that the surface activity results in substances entering the tube at the same rate per unit surface area throughout the length of the tube, a lower quantity of substance would enter the narrow section since the exchange area, πDL, is smaller than in the wide section. Please clarify the discussion in this section supporting the information with the governing equations. 

In general, although I consider that the topic is of scientific importance, some of the mechanisms leading to the blood flow are not sufficiently clear and should be clarified before the accepting the manuscript. 

We thank Reviewer #1 for reviewing our manuscript and for the valuable feedback. We also wish to express our gratitude for bringing up the question about flow direction, as it is likely to be a concern shared by other readers. Addressing this question will therefore enhance the clarity of our work.

To reiterate, the driving force behind surface-induced flow should be the substance-concentration gradient, rather than the absolute quantity of the substance. 

Consider a cylindrical tube with radius R and length L, where substance X can enter or exit the tube through its surface at a rate of amount A per unit surface area per unit time. In this piece of tube, we define the concentration of substance X, denoted as c_X, as the quantity of the substance, n, divided by the volume of the tube, V. Assuming the initial substance concentration is c_(X(t=0)), then at a given time interval ∆t, the concentration of substance c_X can be expressed as the sum of the initial concentration and the change of concentration:

c_X 〖=c_(X(t=0))+∆c〗_X=c_(X(t=0))+∆n/V=c_(X(t=0))+(A∙2πR∙L)/(πR^2∙L) ∆t=c_(X(t=0))+2A∆t 1/R

Considering the model depicted in Fig. 4, where a narrow tube is connected to a wide tube, let us assume that substance X enters the tube (e.g., A is positive) and the initial substance concentration is the same in both tubes. Based on the equation above, at a given time interval ∆t, the concentration difference between the narrow section and the wide section is given by:

〖c_X〗_narrow-〖c_X〗_wide=2A∆t(〖1/R〗_narrow-〖1/R〗_wide )<0

Thus, the substance concentration in the narrower section will be greater than that in the wider section. 

Assuming the flow of the substance in the longitudinal direction is caused by diffusion, the direction of the flow should follow the direction of the concentration gradient - from high concentration to low concentration. Since the substance concentration is higher in the narrow tube, the direction of the longitudinal flow should be from narrow to wide.

We hope the above explanation has cleared up the flow-driving mechanism. We have included the above explanation in the supplemental material.

---

## [Editor Report · Decision Letter 2]

24 Jul 2023

On the driver of blood circulation beyond the heart

PONE-D-22-35747R2

Dear Dr. Li,

We’re pleased to inform you that your manuscript has been judged scientifically suitable for publication and will be formally accepted for publication once it meets all outstanding technical requirements.

Kind regards,

Peter R. Corridon

Academic Editor

PLOS ONE

Additional Editor Comments: Dear authors, I encourage you to provide more details about the experimental system/equipment used in the sections "Dynamics of postmortem blood flow" and "Effect of IR on postmortem & physiological blood flow" to support reproducibility.

---

## [Editor Report · Acceptance letter]

6 Sep 2023

PONE-D-22-35747R2 

On the driver of blood circulation beyond the heart 

Dear Dr. Li:

I'm pleased to inform you that your manuscript has been deemed suitable for publication in PLOS ONE. Congratulations! Your manuscript is now with our production department. 

Kind regards, 

on behalf of

Dr. Peter R. Corridon 

Academic Editor

PLOS ONE